# Risk of motor vehicle collisions after methadone use

Ya-Hui Yang[1], Pei-Shan Ho[2,3], Trong-Neng Wu[4], Peng-Wei Wang[5], Chun-Hung Richard Lin[6], Jui-Hsiu Tsai[7,8,9]*, Yue Leon Guo[8,10]*, Hung-Yi Chuang[8,11,12]*

[1]Department of Health-Business Administration, Fooyin University, Kaohsiung, Taiwan; [2]Division of Medical Statistics and Bioinformatics, Department of Medical Research, Kaohsiung Medical University Hospital, Kaohsiung Medical University, Kaohsiung, Taiwan; [3]Department of Oral Hygiene, College of Dental Medicine, Kaohsiung Medical University, Kaohsiung, Taiwan; [4]Department of Healthcare Administration, Asia University, Taichung, Taiwan; [5]Department of Psychiatry, Kaohsiung Medical University Hospital, Kaohsiung Medical University, Kaohsiung, Taiwan; [6]Department of Computer Science and Engineering, National Sun Yat-sen University, Kaohsiung, Taiwan; [7]Department of Psychiatry, Dalin Tzu Chi Hospital, Buddhist Tzu Chi Medical Foundation, Chia-Yi, Taiwan; [8]Ph.D. Program in Environmental and Occupation Medicine, (Taiwan) National Health Research Institutes and Kaohsiung Medical University, Kaohsiung, Taiwan; [9]Tzu Chi University, Hualien, Taiwan; [10]Environmental and Occupational Medicine, National Taiwan University College of Medicine and NTU Hospital, Taipei, Taiwan; [11]Department of Environmental and Occupational Medicine, Kaohsiung Medical University Hospital, Kaohsiung Medical University, Kaohsiung, Taiwan; [12]Department of Public Health, Kaohsiung Medical University, Kaohsiung, Taiwan

*For correspondence:
faanvangogh@gmail.com (J-HT);
leonguo@ntu.edu.tw (YLG);
ericch@kmu.edu.tw (H-YC)

Competing interests: The authors declare that no competing interests exist.

**Abstract** Methadone maintenance treatment (MMT) can alleviate opioid dependence. However, MMT possibly increases the risk of motor vehicle collisions. The current study investigated preliminary estimation of motor vehicle collision incidence rates. Furthermore, in this population-based retrospective cohort study with frequency-matched controls, opiate adults receiving MMT (cases) and those not receiving MMT (controls) were identified at a 1:2 ratio by linking data from several nationwide administrative registry databases. From 2009 to 2016, the crude incidence rate of motor vehicle collisions was the lowest in the general adult population, followed by that in opiate adults, and it was the highest in adults receiving MMT. The incidence rates of motor vehicle collisions were significantly higher in opiate users receiving MMT than in those not receiving MMT. Kaplan–Meier curves of the incidence of motor vehicle collisions differed significantly between groups, with a significant increased risk during the first 90 days of follow-up. In conclusion, drivers receiving MMT have higher motor vehicle collision risk than those not receiving MMT in opiate users, and it is worthy of noticing road safety in such drivers, particularly during the first 90 days of MMT.

## Introduction

Approximately 58 million people worldwide had opioid user in 2019, with 30 million accounting for opiate users (*United Nations Office on Drugs and Crime, 2020*). Iatrogenic opioid dependence has become an epidemic in many developed countries, particularly the United States (*Anderson, 2017*); in other countries, the number of people using heroin is steadily increasing (*United Nations Office*

**eLife digest** In 2019, 58 million people were estimated to use opioids – a group of substances that include drugs like heroin and morphine. Dependence on opioids can be managed using a prescribed dose of an opioid called methadone, which is administered through a controlled treatment plan. This so-called methadone maintenance treatment manages withdrawal symptoms in opioid-dependent individuals and can reduce the occurrences of overdose, criminal activity and transmission of diseases such as HIV.

However, methadone acts on the same brain receptors as other opioids, and individuals receiving methadone may experience impaired motoric and cognitive functioning, including reduced driving ability. It is therefore important to know whether methadone maintenance treatment may increase an individual's risk to cause road accidents.

To assess motor vehicle collision risk associated with individuals receiving methadone maintenance treatment, Yang et al. analysed data from the Taiwan National Health Insurance Research Database and six Taiwanese administrative registries, including the ministries of health and welfare, interior and justice, and registries in substitution maintenance therapy, road accidents and the National Police Agency.

Initial analyses found that individuals receiving treatment had a higher risk to be involved in car accidents than the general adult population or those without methadone maintenance treatment. Further tests showed that individuals receiving treatment were at three times higher risk of collisions than individuals not receiving treatment, particularly in the first 90 days.

These findings may help individuals undergoing methadone maintenance treatment manage their risk of motor vehicle collisions. Further investigation is needed to reveal the underlying mechanisms of methadone-related impairment of driving ability.

on Drugs and Crime, 2018). Opioid dependence clearly presents a public health challenge worldwide. Methadone maintenance treatment (MMT) is a primary medication-assisted treatment for opioid dependence (United Nations Office on Drugs and Crime, 2020; Darke et al., 2006; Hall et al., 2000; Kleber, 2008). MMT can reduce opioid and heroin dependence, opioid overdose incidence, criminal activity, all-cause mortality, and HIV and hepatitis C virus transmission (Degenhardt et al., 2011; Mathers et al., 2010; Hickman et al., 2018; Kamarulzaman et al., 2016; Chen et al., 2012).

Methadone, a full μ-opioid receptor agonist, can alleviate opioid dependence, both (methadone and opioid) of which can influence psychomotor performance and cognitive functioning in healthy volunteers (Zacny, 1995; Zacny, 1996; Rothenberg et al., 1977; Rothenberg et al., 1980; Rothenberg et al., 1980). Hence, older research described little or no difference in cognitive functioning between MMT patients and healthy controls (Gordon, 1970; Gritz et al., 1975; Appel and Gordon, 1976; Grevert et al., 1977; Appel, 1982; Robinson and Moskowitz, 1985).Given such trends, it is likely that when compared with a higher dose of methadone, MMT patients using a stable dose non-significantly impaired (Moskowitz and Robinson, 1985; Kelley et al., 1978). On the contrary, increasing recent evidence has supported that MMT patients using a stable dose may be impaired on a broad set of neuropsychological tests that related to psychomotor speed, decision-making, working memory, and meta-memory (Mintzer and Stitzer, 2002), information processing, attention, short-term verbal and visual memory, long-term verbal memory and problem-solving (Darke et al., 2000) and cognitive functioning (Mintzer et al., 2005; Verdejo et al., 2005; Prosser et al., 2006; Rapeli et al., 2007; Soyka et al., 2008; Prosser et al., 2009), decision-making (Rotheram-Fuller et al., 2004; Ersche et al., 2006), and driving aptitude (Schindler et al., 2004; Baewert et al., 2007). However, MMT itself may elevate the motor vehicle collision risk (Bramness et al., 2012; Corsenac et al., 2012; Leveille et al., 1994).

Thus far, few observational epidemiological studies (Bramness et al., 2012; Corsenac et al., 2012; Leveille et al., 1994; Babst et al., 1973; Blomberg and Preusser, 1974; Maddux et al., 1977) have been published on the relationship between motor vehicle collisions and MMT; of these, three were performed in the 1970s and used a case-comparison design to investigate drivers receiving MMT in the United States in small-to-medium-sized cohorts (Babst et al., 1973; Blomberg and Preusser, 1974; Maddux et al., 1977). Their results indicated no significant differences in the rate

of motor vehicle collisions between drivers receiving MMT and healthy controls. By contrast, three more recent studies (*Bramness et al., 2012*; *Corsenac et al., 2012*; *Leveille et al., 1994*) found that patients receiving buprenorphine maintenance treatment or MMT had a significantly increased incidence of motor vehicle collisions. Although these studies used medium-to-large-sized cohorts, they neglected some potential risk factors for motor vehicle collisions among drivers receiving MMT, particularly opiate use. Most patients receiving MMT in the aforementioned studies had a history of opioid or heroin dependence. Analgesic opioid users have a 1.8 times higher motor vehicle collision risk than do nonusers (*Leveille et al., 1994*; *Gibson et al., 2009*). Similarly, heroin users have higher motor vehicle collision risk than do healthy people (*Edwards and Quartaro, 1978*). Thus, opiate use history must be identified when estimating motor vehicle collision risks related to MMT use. Few studies have proposed effective measures for motor vehicle collision prevention in drivers receiving MMT.

To investigate whether drivers receiving MMT have an increased motor vehicle collision risk, we analyzed nationwide motor vehicle collision incidence rate in three groups as preliminary data: general adult population, adult opiate users, and adults receiving MMT. Furthermore, by using these data, we created population-based matched retrospective cohorts of new opiate users receiving and not receiving MMT. Moreover, we should provide some suggestions for early prevention of motor vehicle collisions in opiate users receiving MMT.

## Methods

### Data source

Data were retrieved from the Taiwan National Health Insurance Research Database (NHIRD) (*Wu and Lee, 2016*; *Hu et al., 2019*) and six Taiwanese population-based administrative registries, namely the management information system of substitution maintenance therapy, Ministry of Health and Welfare, the road accident registry of injurious crashes, National Police Agency, Ministry of the Interior, and four independent management information systems at the Ministry of Justice, Republic of China (Taiwan). The four information systems were the case management system of Drug Prevention and Control Center, processing system of criminal records, criminal case system of drug case prosecutor briefed the transfer of information, and punitive administrative system for the use of Category 3 or 4 Narcotics (illicit drugs). Preliminary data were independently collected by the aforementioned governmental departments and managed by the Health and Welfare Data Science Center, Ministry of Health and Welfare. Data from different systems were linked using the unique national identification numbers assigned to each citizen in Taiwan. For the consideration of privacy protection, all of the personal identifications were recorded, only authorized researchers were permitted to process databases in a separated designate area, and only statistical results were allowed to be carried out for publications. Personal identifiers were removed after the linkage and before the analysis.

### Study population

We combined and organized the four registry databases from the four independent management information systems at the Ministry of Justice (*Figure 1*). The total number of opiate users in the registry from 1956 to 2016 was 107,213. From the four registry databases, we identified new opiate users between 2010 and 2016 (n = 15,996), who were defined the first detection by law enforcement. The new opiate users were excluded if they (1) were registered at <20 years of age; (2) used opiate before 2010; and (3) had incomplete information on age, sex, education status, income, residing in area, etc. Of the new opiate users, we selected those receiving MMT as the (MMT) exposed group, who were regular methadone users. The date of first MMT administration was defined as the index date. Other new opiate users not receiving MMT were randomly selected as the (MMT) unexposed group after they were frequency-matched to the exposed group at a ratio of 1:2 according to age, sex, and opiate use duration. Thereafter, the index date of two matched unexposed users was the same day as that of the exposed user. The included participants had not been in jail after their index date. *Figure 1* depicts the flow of patient selection in the present study. Potential covariates, including history of motor vehicle collisions, driving under the influence (DUI), antidepressant use, and BZD (including Z-drug) use before the index date, were included in the analysis

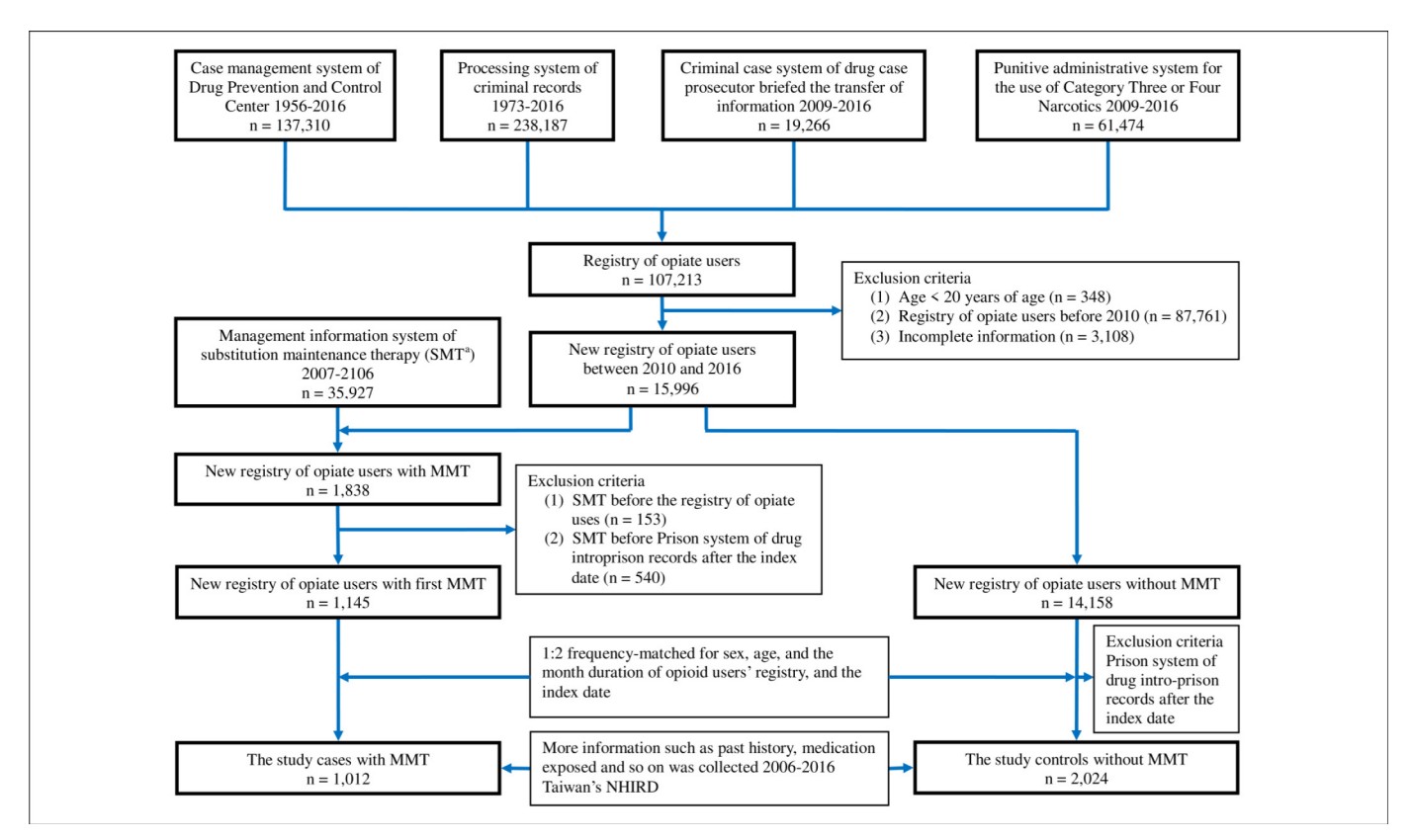

**Figure 1.** Flow chart of participant selection.

(*Bramness et al., 2012*; *Corsenac et al., 2012*; *World Health Organization, 2018*; *Chang et al., 2013*; *Engeland et al., 2007*; *World Health Organization, 2004*).

## Study outcomes and potential covariates

The road accident registry of injurious crashes provided information regarding motor vehicle collisions involving personal death, injury, or vehicle damage on Taiwanese roads; the subjects of the current study were focused on the drivers of the road accidents in the database. The main outcome was the incidence of motor vehicle collisions after the index date. All participants were followed until motor vehicle collision after the index date, death, end of follow-up in registry records, or the end of 2016.

Data regarding methadone treatment were extracted from the 2007–2016 management information system of substitution maintenance therapy, which includes information on all prescriptions issued by at least two psychiatrists in Taiwan. This registry omits drug administration information of individuals who were hospitalized or received medications dispensed by outpatient departments. Methadone is dispensed to individuals who meet the criteria for opioid use (dependence or abuse) as defined by the *International Classification of Disease, Ninth Revision, Clinical Modification* (*ICD-9-CM* 304.00–304.03, 304.70–304.83, and 305.50–305.53). Patients receiving MMT must comply with daily witnessed ingestion under the supervision of a pharmacist or psychiatric nurse and are forbidden to take medication away from treatment sites. Methadone dosing and treatment duration are individualized, varying according to patient tolerance and clinical response across treatment stages (induction, titration, and stabilization) according to the Regulations for MMT Guidelines in *Taiwan Centers for Disease Control, 2007*; *Department of Health, Executive Yuan of Taiwan, 2006*.

## Statistical analyses

All statistical analyses were performed using SAS (version 9.1, SAS Institute, Cary, NC, USA). Participants were stratified on the basis of age, sex, duration of opiate use, education status, income level, urbanity, history of motor vehicle collisions, DUI, antidepressant use, and BZD (Z-drug) use by using the Pearson $\chi^2$ test. To determine the independent effect of MMT on motor vehicle collision risk in a previous opiate user, we used a proportional hazard model after adjusting for motor vehicle collisions, as stated previously. The Kaplan–Meier method and log-rank test were used to estimate and compare the incidence of motor vehicle collisions between participants receiving and not receiving MMT, assuming a two-tailed alpha level of statistical significance of 0.05.

## Results

### Crude incidence rates of motor vehicle collisions

*Figure 2* shows the crude incidence rates (CIRs) of motor vehicle collisions in Taiwan during 2009 to 2016 among the general adult population, adult opiate users, and adults receiving MMT. The CIRs of motor vehicle collisions in the general adult population slightly increased from 19.2 per 1000 person-years in 2009 to a peak of 30.6 per 1000 person-years in 2014 but steadily decreased to 30.3 per 1000 person years in 2016. Over the follow-up period, the CIRs of motor vehicle collisions in adultopiate users followed a similar trend—from 28.2 to 46.8 per 1000 person-years. In the general adult population receiving MMT, the highest CIR of motor vehicle collisions was noted in 2012 (58.7 per 1000 person-years), with a wide range of 37.5 to 58.7 per 1000 person-years. Overall, the CIRs of motor vehicle collisions from 2009 to 2016 were the lowest in the general adult population, followed by those in adult opiate users, and they were the highest in adults receiving MMT.

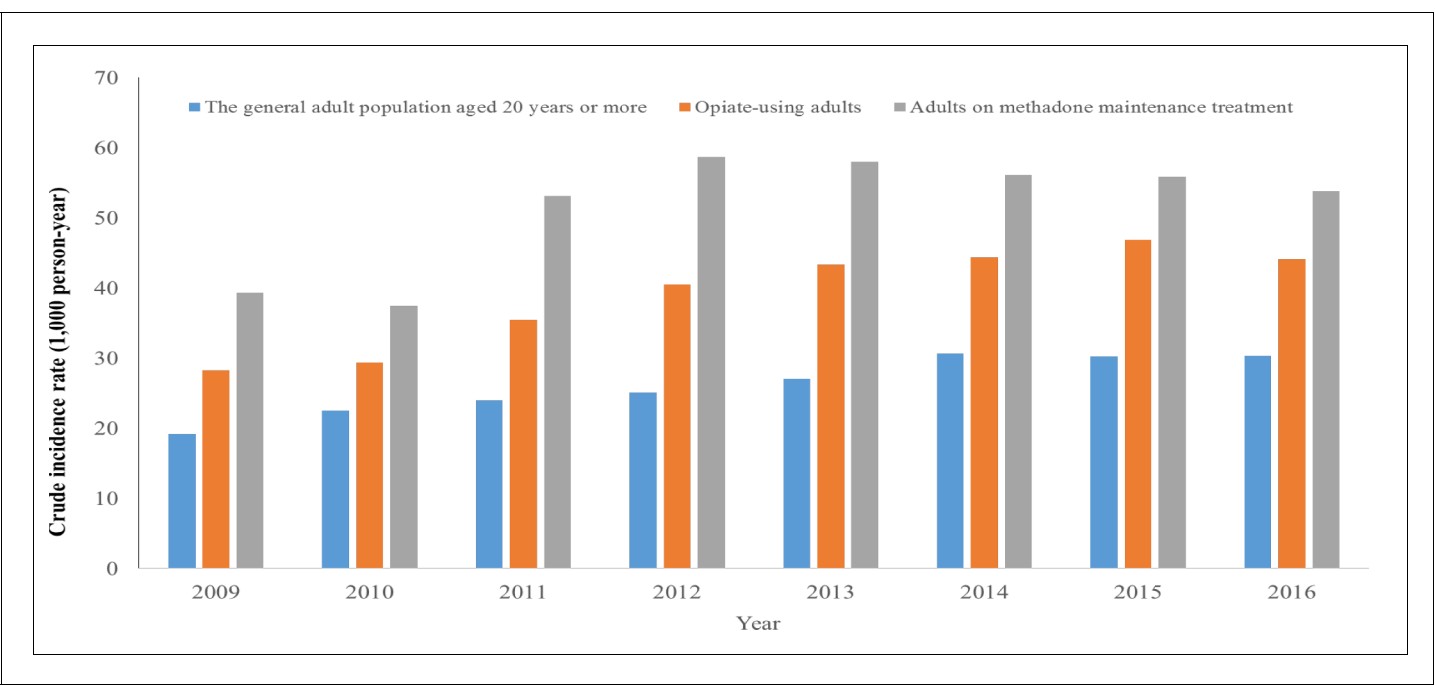

**Figure 2.** Crude incidence rates (CIRs) of motor vehicle collisions annually among the general adult population, adult opiate users, and patients receiving methadone maintenance treatment (MMT). The CIRs of motor vehicle collisions in Taiwan during 2009–2016 among the general adult population, adult opiate users, and adults receiving MMT are shown. The CIRs of motor vehicle collisions in the general adult population slightly increased from 19.2 per 1000 person-years in 2009 to a peak of 30.6 per 1000 person-years in 2014 but steadily decreased to 30.3 per 1000 person-years in 2016. Over the follow-up period, the CIRs of motor vehicle collisions in adult opiate users followed a similar trend—from 28.2 to 46.8 per 1000 person-years. In the general adult population receiving MMT, the highest CIR of motor vehicle collisions was noted in 2012 (58.7 per 1000 person-years), with a wide range of 37.5–58.7 per 1000 person-years. Overall, the CIRs of motor vehicle collisions from 2009 to 2016 were the lowest in the general adult population, followed by those in adult opiate users, and they were the highest in adults receiving MMT.

## Cohort population characteristics

Data from 3036 opiate users—including 1012 opiate users receiving MMT (MMT group) and 2024 opiate users not receiving MMT (controls)—were included in the 7-year follow-up cohort (median 5.0 years per person; interquartile range 2.9–5.9 years). The baseline characteristics of opiate users are shown in *Table 1*. The mean age (± standard deviation) at presentation was 37.7 (± 8.1) years. The mean duration of opiate use was 15.3 (± 15.4) months, and 85.6% (n = 2598) of the participants were male. No significant between-group differences were observed in these characteristics, including urbanization of the area of residence and history of motor vehicle collisions and DUI. The participants receiving MMT had a lower education level but higher income than did the controls (both p<0.001). The MMT group had significantly more exposure to antidepressants (8.4 and 5.1%, respectively; p<0.001) and BZD (Z-drugs) (63.5 and 18.2%, respectively; p<0.001) than did the controls.

## Study outcomes and potential covariates

During the study period, the incidence rates of motor vehicle collisions were 6.5 and 2.2% in the MMT group and controls, respectively. The mean time interval from the index date to a motor vehicle collision was similar between the groups (234.8 ± 295.0 and 226.7 ± 287.7 days, respectively; p=0.886; *Table 1*). *Figure 3* shows the Kaplan–Meier analysis for motor vehicle collision-free survival between opiate users receiving MMT and those not receiving in long-term and short-term follow-up. From the survival curve of 7-year follow-up, the upper panel of *Figure 3* shows that both MMT and

**Table 1.** Baseline characteristics of opiate users in Taiwan, 2010–2016.

| Characteristics | MMT n = 1012 | | Non-MMT n = 2024 | | p |
|---|---|---|---|---|---|
| Age, mean (SE), years | 37.7 (8.1) | | 37.7 (8.1) | | 1.000 |
| Length of time on opiate users' registry, mean (SE), months | 15.3 (15.4) | | 15.3 (15.4) | | 1.000 |
| Sex, n (%) | | | | | 1.000 |
| Male | 866 | (85.6) | 1732 | (85.6) | |
| Female | 146 | (14.4) | 292 | (14.4) | |
| Education status, n (%) | | | | | <0.001 |
| Elementary school (1–6 years) | 91 | (9.0) | 115 | (5.7) | |
| High school (7–12 years) | 896 | (88.5) | 1832 | (90.5) | |
| College or more (>12 years) | 25 | (2.5) | 77 | (3.8) | |
| Income level, n (%) | | | | | <0.001 |
| ≦15,000 (New Taiwan $) | 615 | (60.8) | 1394 | (68.9) | |
| Urbanity, n (%) | | | | | 0.240 |
| Urban | 701 | (69.3) | 1453 | (71.8) | |
| Suburban | 27 | (2.7) | 60 | (3.0) | |
| Rural | 284 | (28.1) | 511 | (25.3) | |
| Past history of, *n* (%) | | | | | |
| Motor vehicle collision | 339 | (33.5) | 623 | (30.8) | 0.129 |
| Driving under the influence | 67 | (6.6) | 106 | (5.2) | 0.121 |
| Antidepressant use | 85 | (8.4) | 104 | (5.1) | <0.001 |
| Benzodiazepine (Z-drug) use | 643 | (63.5) | 369 | (18.2) | <0.001 |
| Occurrence of | | | | | |
| Motor vehicle collision, *n* (%) | 66 | (6.5) | 45 | (2.2) | <0.001 |
| Duration*, mean (SE), days | 234.8 (295.0) | | 226.7 (287.7) | | 0.886 |

*Length of time from the index date to motor vehicle collision event after the index date, death, end of follow-up in registry, or the end of 2016.
MMT: methadone maintenance treatment.

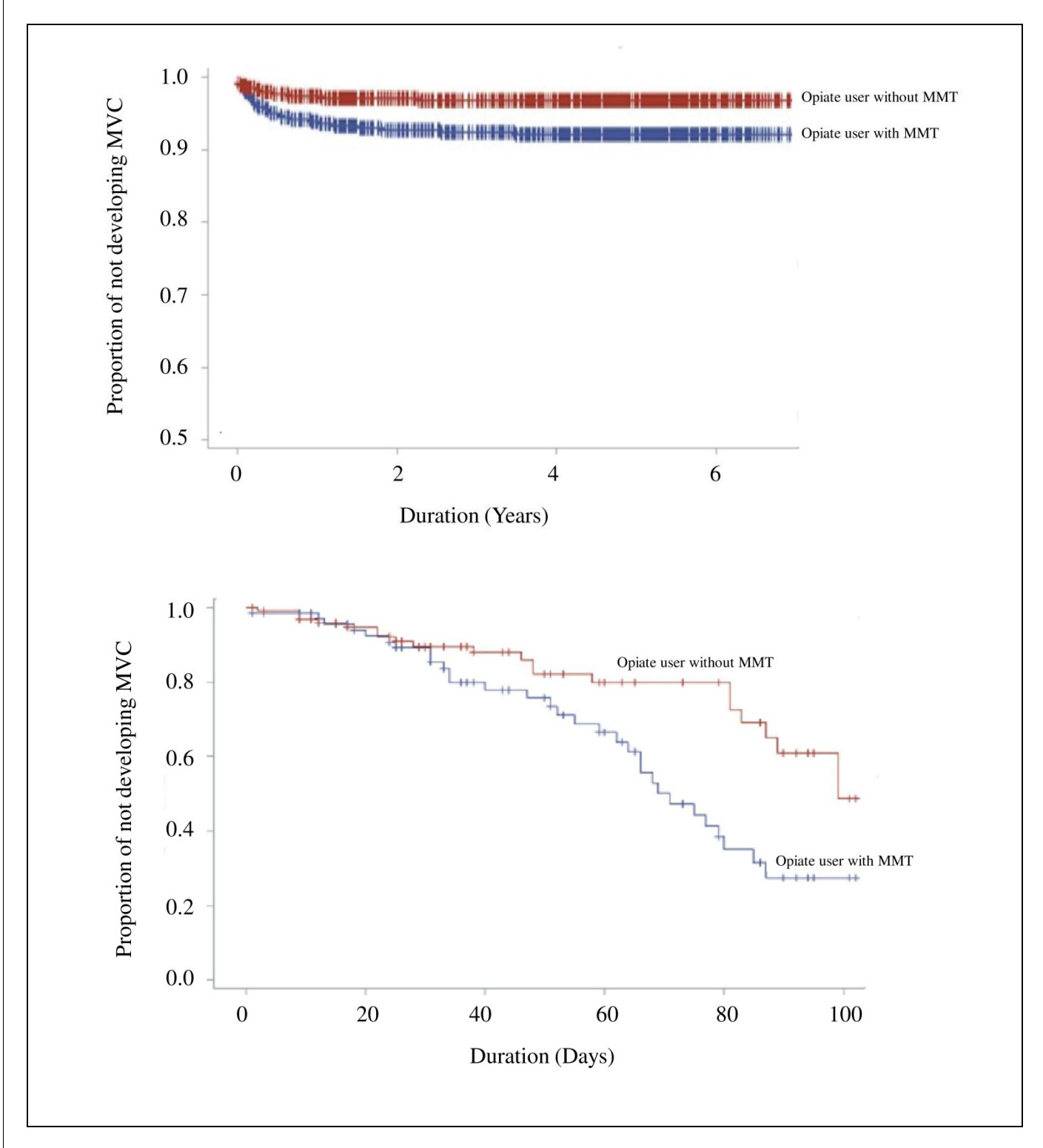

**Figure 3.** The Kaplan–Meier curve of motor vehicle collision-free between opiate users receiving methadone maintenance treatment and those not receiving in long-term (7 years, upper panel) and short-term (100 days, lower panel) follow-up.

control groups had similar patterns of motor vehicle collision-free survival. From the further analysis during 100-day follow-up, it began to have no differences between both groups in the first 30 days of MMT intervention, but, after the first 30 days, differences were noted. Notably, a rapid descending curve in the MMT group was discovered in the first 90 days; later, the curve descended steadily. Overall, descending rate of motor vehicle collision-free survival in the MMT group (6.5%; range 5.23–7.08%; 95% confidence interval [95% CI], 5.23–7.08) was significantly higher than that in the control group (2.2%; range 1.90–2.41%; log-rank test p<0.001) (*Figure 3*).

Univariate analysis results indicated that compared with the controls, opiate users receiving MMT had an increased risk of motor vehicle collisions (crude hazard ratio [HR] 3.00 [95% CI, 2.05–4.38]; log-rank test p<0.001). The risk of motor vehicle collisions was the highest in participants with a history of DUI (crude HR 2.26 [95% CI, 1.26–4.01]; p=0.006), followed by that in those with antidepressant exposure (crude HR 2.25 [95% CI, 1.26–3.93]; p=0.005). Factors predictive of motor vehicle collisions were history of motor vehicle collisions (crude HR 1.70 [95% CR, 1.17–2.48]; p=0.006), BZD (Z-drug) use (crude HR 1.62 [95% CI, 1.09–2.42]; p=0.018), and rural location (crude HR 1.68 [95% CI, 1.14–2.46]; p=0.009; *Table 2*). Multivariate analysis showed that after adjustments for income level, urbanity, education status, history of motor vehicle collisions, DUI, BZD (Z-drug) use, and antidepressant use, the adjusted HR for motor vehicle collisions in the MMT group was 2.75 (95% HR, 1.87–4.04; p<0.001), indicating that opiate users receiving MMT had a significantly increased motor vehicle collision risk. In addition, participants residing in urban areas were 1.56 times more likely to encounter motor vehicle collisions (95% CI, 1.05–2.32; p=0.027) than were controls (after adjustment; *Table 2*). No differences were observed in the incidence of motor vehicle collisions between groups with respect to history of motor vehicle collisions (adjusted HR 1.41 [95% CI, 0.94–2.10]), DUI (adjusted HR 0.61 [95% CI, 0.33–1.13]), antidepressant use (adjusted HR 1.70 [95% CI, 0.95–3.05]), or BZD (Z-drug) use (adjusted HR 1.25 [95% CI, 0.82–1.91]).

## Discussion

From 2009 to 2016, the crude incidence rates (CIRs) of motor vehicle collisions were the lowest in the general adult population, followed by those in adult opiate users, and the highest in adults

**Table 2.** Independent predictors of motor vehicle collisions among opiate users receiving methadone maintenance treatment (MMT).

| Variables | | Crude hazard ratio (95% CI) | p | Adjusted hazard ratio (95% CI) | p |
|---|---|---|---|---|---|
| Education status | | | | | |
| | Elementary school | 1.00 | - | 1.00 | - |
| | High school | 0.90 (0.50, 1.64) | 0.727 | 1.06 (0.53, 2.12) | 0.875 |
| | College or more | 0.83 (0.26, 2.60) | 0.743 | 1.05 (0.28, 3.95) | 0.942 |
| Income level (New Taiwan $) | | | | | |
| | ≦15,000 | 1.27 (0.87, 1.86) | 0.220 | 1.14 (0.77, 1.67) | 0.516 |
| Urbanity | | | | | |
| | Urban | 1.00 | - | 1.00 | - |
| | Suburban | 0.62 (0.15, 2.50) | 0.500 | 0.76 (0.19, 3.10) | 0.700 |
| | Rural | 1.68 (1.14, 2.46) | 0.009 | 1.56 (1.05, 2.32) | 0.027 |
| Past history of | | | | | |
| | Motor vehicle collision (vs. no) | 1.70 (1.17, 2.48) | 0.006 | 1.41 (0.94, 2.10) | 0.097 |
| | Driving under the influence (vs. no) | 2.26 (1.26, 4.01) | 0.006 | 0.61 (0.33, 1.13) | 0.116 |
| | Antidepressant use (vs. no) | 2.25 (1.28, 3.93) | 0.005 | 1.70 (0.95, 3.05) | 0.076 |
| | Benzodiazepine (Z-drug) use (vs. no) | 1.62 (1.09, 2.42) | 0.018 | 1.25 (0.82, 1.91) | 0.296 |
| Occurrence of motor vehicle collision | | | | | |
| | MMT (vs. non-MMT) | 3.00 (2.05, 4.38) | <0.001 | 2.75 (1.87, 4.04) | <0.001 |

receiving MMT. In the 7-year cohort study with frequency-matched controls, the motor vehicle collision incidence rate among opiate users was significantly higher in those receiving MMT than in those not receiving MMT. This rate drastically increased in opiate users receiving MMT during the first 90 days of follow-up. Rural area residents had increased motor vehicle collision risk. Although the occurrences of motor vehicle collisions in drivers under the influence of current opiate use, current opiate use during opiate withdrawal, BZD, and alcohol are higher than those receiving MMT, these factors (such as the history of BZD, antidepressants, and alcohol exposed) were nonsignificantly increased risk of motor vehicle collisions in opiate users receiving MMT.

In the preliminary analysis based on the nationwide CIRs of motor vehicle collisions, the CIRs in the general and opiate-using adult populations slowly increased toward a peak, followed by a slight decline. By contrast, the CIR in adults receiving MMT remained high, without such fluctuation. The risk of motor vehicle collisions was 1.45–1.62 times higher in adult opiate users and 1.66–2.34 times higher in adults receiving MMT than in the general adult population. Thus, adults receiving MMT had a consistently higher rate of motor vehicle collisions than did the general adult population. These findings corroborate those of three observational studies (*Bramness et al., 2012*; *Corsenac et al., 2012*; *Leveille et al., 1994*). These results do not eliminate the effects of interfering factors such as opiate use because most of the participants used opiates before receiving MMT. Furthermore, we included a retrospective cohort of opiate users receiving and not receiving MMT to eliminate such effects.

In this cohort study, among 3036 opiate users, those receiving MMT had a significantly higher motor vehicle collision risk than did those without such treatment. The percentage of opiate users with a motor vehicle collision event was approximately three times higher in the participants receiving MMT than in those not receiving MMT (6.5% vs. 2.2%). Compared with opiate users not receiving MMT, those receiving MMT had an increased risk of BZD (Z-drug) or antidepressant use, a result consistent with those of previous studies (*Bramness et al., 2012*; *Corsenac et al., 2012*; *Bramness and Kornør, 2007*). Univariate analysis showed that factors predictive of motor vehicle collisions in opiate users included a history of DUI, antidepressant use, BZD (Z-drug) use, and motor vehicle collisions. We found no significant difference in the incidence rate of motor vehicle collisions between the opiate user groups with respect to history of motor vehicle collisions, DUI, antidepressant use, and BZD (Z-drug) use. These findings suggest that these risk factors for motor vehicle collisions in the general population may nonsignificantly affect motor vehicle collisions in adult opiate users receiving MMT. Our results corroborate those of previous studies, demonstrating a positive correlation between motor vehicle collision risk and drivers' exposure to methadone (*Bramness et al., 2012*; *Corsenac et al., 2012*; *Leveille et al., 1994*). In addition, our findings underscore the importance of considering motor vehicle collision risk during the first 90 days of methadone treatment. Authorities, physicians, and methadone users and their families should be educated regarding the elevated risk to prevent motor vehicle collisions. Further studies are needed to determine whether a policy forbidding driving during the first 90 days of MMT could effectively reduce motor vehicle collision risk.

The cause underlying the correlation between motor vehicle collisions and MMT use remains controversial. Studies have shown that methadone causes no or only slight impairment of psychomotor performance, particularly in chronic MMT (*Specka et al., 2000*; *Brown et al., 2004*; *Rothenberg et al., 1977*; *Curran et al., 2001*). By contrast, other studies have shown that methadone-maintained patients experience cognitive deficiencies (*Specka et al., 2000*; *Staak et al., 1993*; *Dittert et al., 1999*; *Darke et al., 2000*; *Mintzer et al., 2005*). Moreover, following 6 months of MMT compared to the first month, improved visuospatial construction and executive function, but no change in memory or attention performance, were found in a cross-sectional study (*Soyka et al., 2010*). Our results clearly demonstrate an elevated risk of motor vehicle collisions in drivers receiving MMT, particularly during the first 90 days of treatment. One possible explanation for this result is that MMT causes cognitive impairment only during the early stages of treatment.

## Strengths and limitations

The major strength of our study is the use of a population-based sample of nationwide motor vehicle collision incidences and a case-comparison cohort for motor vehicle collisions. The findings may provide the basis for designing measures to prevent motor vehicle collisions during MTT, particularly in ethnic Chinese populations. Most observational studies showed different risks of motor vehicle

collisions between two populations (the general population and the MMT population). We not only added the third population (the opiate-using population) in the preliminary study but also controlled for the potentially contributing factor of opioid use in the cohort study of opiate users receiving MMT and those not receiving. Furthermore, the data are not subject to reporting or recall bias because we used high-quality data from six administrative registries and the NHIRD. Nevertheless, we acknowledge several limitations. First, the administrative data was the lack of information on road exposure (driven hours or distances in a given period) because unobservable differences between the road exposure may confound the relationship of interest. Motor vehicle collisions in the study were assessed using the police registry. Hence, unreported or minor motor vehicle collisions may not be included in our analysis. Second, the number of opiate users and duration of opiate use may be underestimated. Opiate use is often underreported because it is illegal in Taiwan. Moreover, opioid doses were recognized much higher risks among MMT users than among non-MMT opioid users; hence, opiate dose was uncaptured because it was unavailable in our database. The data about enrollment in a nongovernmental organization institution for rehabilitation and severity of opiate use were lacking. Third, data from drivers' licenses were lacking in our registry. Although motor vehicle collision risk may be underestimated in the study, we excluded participants aged <20 years to diminish underestimation. Fourth, the degree of exposure to unfavorable road conditions and information regarding unsafe road infrastructure, inadequate traffic laws, car speed, driver fatigue, talking on cell phones, and unsafe vehicles was unavailable in the present study (*World Health Organization, 2018*; *Chang et al., 2013*; *Engeland et al., 2007*; *World Health Organization, 2004*). Finally, the NHIRD only provides information regarding the dispensing of prescribed medications. Because nonadherence is considered a potential confounder, caution should be exercised when comparing our findings with the results reported by other groups in which data are collected from clinical settings (*Chang et al., 2013*; *Engeland et al., 2007*; *World Health Organization, 2004*; *Babst et al., 1973*; *Blomberg and Preusser, 1974*; *Maddux et al., 1977*; *Gibson et al., 2009*; *Edwards and Quartaro, 1978*).

In line with previous findings, we provided compelling evidence that opiate users on MTT have a significantly increased motor vehicle collision risk. Individuals receiving MMT should be informed of this risk, so that they can take appropriate measures to prevent motor vehicle collisions, particularly during the first 90 days of MMT and if living in rural areas.

## Acknowledgements

We would like to thank Miss Mei-Hsin Ho for her assistance and reference management. The funders had no role in study design, data collection, data analysis, data interpretation, or writing of the report. Some of the data used in the present study were sourced from the Health and Welfare Data Science Center, Ministry of Health and Welfare (registered nos.: H106038 and H107088).

## Additional information

### Funding

| Funder | Grant reference number | Author |
|---|---|---|
| Taiwan Food and Drug Administration | MOHW106-FDA-D-114-000671 | Ya-Hui Yang |
| Taiwan Food and Drug Administration | MOHW107-FDA-D-114-000636 | Ya-Hui Yang |

The funders had no role in study design, data collection and interpretation, or the decision to submit the work for publication.

### Author contributions

Ya-Hui Yang, Conceptualization, Data curation, Formal analysis, Supervision, Methodology, Writing - original draft, Project administration; Pei-Shan Ho, Formal analysis, Methodology, Writing - review and editing; Trong-Neng Wu, Supervision, Writing - review and editing; Peng-Wei Wang, Data curation, Project administration, Writing - review and editing; Chun-Hung Richard Lin, Yue Leon Guo,

Formal analysis, Writing - review and editing; Jui-Hsiu Tsai, Conceptualization, Data curation, Funding acquisition, Methodology, Writing - original draft, Writing - review and editing; Hung-Yi Chuang, Formal analysis, Project administration, Writing - review and editing

#### Author ORCIDs

Ya-Hui Yang (ID) https://orcid.org/0000-0002-1785-5123
Pei-Shan Ho (ID) https://orcid.org/0000-0002-0860-1690
Jui-Hsiu Tsai (ID) https://orcid.org/0000-0001-7335-4131
Yue Leon Guo (ID) https://orcid.org/0000-0002-8530-4809
Hung-Yi Chuang (ID) https://orcid.org/0000-0002-8321-8720

#### Ethics

Human subjects: The study was approved by the Institutional Review Boards (TSMH IRB No.: 17-010-B1 and 18-049-B); the need for written informed consent was waived because we used deidentified information from the databases.

#### Decision letter and Author response

Decision letter https://doi.org/10.7554/eLife.63954.sa1
Author response https://doi.org/10.7554/eLife.63954.sa2

## Additional files

#### Supplementary files

• Transparent reporting form

#### Data availability

Data are available from the National Health Insurance Research Database (NHIRD) published by Taiwan National Health Insurance (NHI) Bureau. Due to legal restrictions imposed by the government of Taiwan in relation to the "Personal Information Protection Act", data cannot be made publicly available. Requests for data can be sent as a formal proposal to the NHIRD (http://nhird.nhri.org.tw).

The following datasets were generated:

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
