## [Decision Letter]

**Acceptance summary:**

This work provides important real-world context for the impact of methadone maintenance therapy for opioid use disorder. The results indicate that rates of motor vehicle collisions are increased in adults on methadone maintenance compared even with opiate users not on methadone maintenance. This study is of growing importance as the opioid misuse crisis continues world-wide and as methadone maintenance therapy is made more broadly available to a growing population with opioid use disorders.

**Decision letter after peer review:**

Thank you for submitting your article "Risk of motor vehicle accidents after methadone use" for consideration by *eLife*. Your article has been reviewed by 3 peer reviewers, including Michael Taffe as the Reviewing Editor and Reviewer #1, and the evaluation has been overseen by Timothy Behrens as the Senior Editor.

Essential revisions:

1) The methods are not described sufficiently for a full evaluation of the study. It is essential to address all the comments made by the reviewers regarding how the subjects were selected, matched, defined, etc. These characteristics of the sample then need to be addressed in the Discussion.

2) The manuscript does not go far enough in addressing the potential impact of intoxicants, of severity of OUD, of trying to account for accident rates in active users, etc. Additional methodological details and correlates such as the time of day of the accidents may address some of these concerns.

3) The conclusions should be carefully contextualized in terms of the magnitude of any risks, the context of enhanced rates of accidents, the risks of untreated OUD patients, etc. These results are likely to inform MMT practices and as such the manuscript should discuss these implications thoughtfully and carefully.

*Reviewer #1:*

Tsai and colleagues report a study of the rate of motor vehicle accidents in populations of opioid users on methadone maintenance therapy (MMT) versus those not on MMT and matched control non-opioid users. In the context of the ongoing opioid use crisis in many countries around the world, and the growing efforts to treat users with methadone maintenance therapy (MMT), it is critical to understand the risks and benefits of such therapies. This study addresses motor vehicle accident rates in opioid users in MMT as compared with the general adult population but also includes assessment of opioid users who are not on MMT. This inclusion is a key advance of understanding the risks of MMT. The study captures national statistics in Taiwan for a 7 year period (2009-2016) which include 3036 opioid users of which 1012 were in MMT. The first approach was to compare the incidence of motor vehicle accidents in each population in each year of the data interval. These data show the highest rates in MMT users and the lowest in the general population for each year of study. Most importantly, by tracking individuals after an index date of the first MMT dose in this group, and a matched date for the non-MMT controls, this study shows the cumulative probability of a motor vehicle accident increases more rapidly in the MMT opioid group versus the non-MMT opioid group in the first 100 days after the index date.

Major strengths of this study include the capture of a large number of individuals through national databases, the comparison of MMT and non-MMT opioid users groups and the determination of the latency of motor vehicle accidents after the initiation of therapy. Minor strengths are captured by the inclusion of covariates such as a prior history of motor vehicle accidents, driving under the influence and benzodiazepine use.

One major limitation of this study, as presented, lies in a lack of clarity about the opioid use history of the subjects as the approach only captures their first interaction with law enforcement. Relatedly, the manuscript is unclear about the matching of non-MMT and MMT subjects in terms of their opioid use history and other characteristics. The Discussion does not address this with any indication of the factors in Taiwan that determine whether an opioid users is placed on MMT or not, particularly with respect to opioid use intensity and longevity of the pattern, and does not address how this may have affected the results. Minor limitations of the study are that the MMT group had significantly lower education level and more exposure to benzodiazepines and less to antidepressants compared with the non-MMT control group, but the study does not describe the likely magnitude of the contribution of these factors to accident rates. The study has largely achieved the aims of the authors by investigating the risks posed by MMT for motor vehicle accidents. The results presented confirm increased risk and show that it develops over the first 100 days. These results potentially contribute to reduce public harm by suggesting increased caution for MMT patients in the several months after initiation of therapy.

1. Results, Page 11: The final paragraph is confusing, perhaps because it is not well described in the methods or the results that this analysis (and that in the lower panel of Figure 3) focuses on the sub-populations who did eventually get in a motor vehicle accident. This is not directly stated anywhere so if this is not true, then it is even less clear what is being depicted in Figure 3.

2. Figure 3: It is unclear from the legend and the labels on the axes what is being shown in the upper panel of Figure 3. The legend should be revised to make this very clear.

3. Discussion Page 13: "this rate declined drastically…during the first 90 days." This appears to be referring to the lower panel of Figure 3 in which the cumulative rate of motor vehicle accidents *increased* over the first 90 days. This all needs to be better clarified, particularly with respect to the year intervals reported in Figure 3, upper panel.

4. Discussion Page 14: It appears from the Introduction comment that the "3 observational studies" cited did not feature a non-MMT group with an opioid use history? If that is the case it should be re-emphasized here since this appears to be a strength of this manuscript.

5. Discussion Page 16: Although it is appreciated that road conditions and driver behavior are not captured in the database, it may be that the time of day and day of the week when accidents occurred would be useful variables, and should be assessed.

6. Methods, Page 19: The authors should address the factors in Taiwan that determine whether an opioid users is placed on MMT or not, particularly with respect to opioid use intensity and longevity of the pattern.

7. Methods, Page 19: The selection of the "index date" for the non-MMT needs to be better described. Would it be possible to include an entry in the Table to describe the latency between the placement on the opioid registry and the index date for each group?

8. Methods, Page 19: It should be stated directly that the subject was the driver of the car in the motor vehicle accidents, to avoid any confusion.

9. Methods, Page 19: "on the basis of age, sex, and opiate use duration". This appears to suggest a design in which the non-MMT users were selected by matching to the MMT users but this needs to be better clarified.

10. Methods, Page 18-19: It appears to be the case that all opioid users would have to have a conviction for use of illicit drugs. This should be clarified, particularly with respect to the description as "new" opioid users. Is this defined by their first interaction with / detection by law enforcement? Or was there some sort of interview to assess when they started using opioids?

*Reviewer #2:*

1) Data were retrieved from NHRID, and 6 Taiwanese population-based administrative registries, but the 6 population-based administrative registries may high repeat and overlapping such as 1. management information system of substitution maintenance therapy belong of Ministry of Health and Welfare, 2. National Police Agency belong of Ministry of the Interior, 3. Should be clarify the MVA information of road accident registry of injurious crashes from Ministry of Transport or National Police Agency.

2) The Methadone is a long action medicine, which induce motor vehicle accident less than patient with current OUD/or current OUD during opiate withdrawal, and less than current using BZD or alcohol. This study has not clear clarification these influence factors.

3. The study conclusion is too arbitrary, and suggested that receiving MMT have high motor vehicle accident (MVA) risk than not receiving MMT in patient with opioid use disorder (OUD), and should prohibition against driving during MMT. The major risk of driving is drinking driver and drug (opiate and BZD) driver, the study could not strong to excluded that multi-factory risks in MVA. The treatment with MMT play an important successful role for control HIV infection in Taiwan patient with OUD, and OUD patient must regular take methadone every day.

4. Opiate users not receiving MMT may be just discharged from jail or enrolled in a NGO institution for rehabilitation, or their current severity of opiate use may not met the OUD criteria, or the current severity in non-MMT group may less than receiving MMT group (because received MMT case must met the criteria of OUD, and most case presents in severe degree). In addition, persistent opiate user may more high risk of MVA and other accident if not receiving any treatment, because persistent current opiate (heroin) user shows high craving for drug, or presents in withdrawal state. The high craving ad withdrawal state are all dangerous state for anything (including MVA and many harm accident).

5. The Methadone is a long action medicine, which induce motor vehicle accident less than patient with current OUD/or current OUD during opiate withdrawal, and less than current using BZD or alcohol. These factors should clarification in the conclusion.

6. The data of UNODC using 2018 is too old, the UNODC data of 2020 is available.

Method:

7. Data were retrieved from NHRID, and 6 Taiwanese population-based administrative registries, but the 6 population-based administrative registries may high repeat and overlapping such as 1. management information system of substitution maintenance therapy belong of Ministry of Health and Welfare, 2. National Police Agency belong of Ministry of the Interior, 3. Should be clarify the MVA information of road accident registry of injurious crashes from Ministry of Transport or National Police Agency.

8. The Methadone is a long action medicine, which induce motor vehicle accident less than patient with current OUD/or current OUD during opiate withdrawal, and less than current using BZD or alcohol. This study has not clear clarification these influence factors.

*Reviewer #3:*

As I understand the study methods, the researchers used six linked Taiwanese national administrative datasets to identify 1012 adults initiating MMT between 2009 and 2016, matching these individuals on the basis of age, sex, and 'opiate use duration' to a control group of 2024 new users of non-methadone opioids. Individuals were eligible for inclusion if they were new users of opioids between 2010 and 2016. Individuals were excluded if they: were registered as new opioid users prior to age 20 years; used opioids prior to 2010 (I believe the washout period was 2007 – 2009); had incomplete information on age, sex, or "multiplicity" (I did not understand the final term and wonder if the authors intended to use another word); or if they were subsequently incarcerated.

The index date for the MMT group was the date of first MMT administration. I had difficulty understanding the index date for the control group. The primary outcome of interest was subsequent police-reported motor vehicle collision (MVC). Individuals were followed until MVC, death, end of follow-up in registry records, or end of study interval.

Using a proportional hazards model to assess MVC-free survival, researchers report that the hazards of MVC were significantly higher among MMT users than among controls (crude hazard ratio, 3.0). After adjustment for income level, urbanity, education status, history of motor vehicle accidents, prior alcohol-impaired driving, prescription Z-drug exposure, and antidepressant use, researchers report that the adjusted hazard ratio for MVC was 2.8-fold higher among individuals receiving MMT relative to individuals receiving non-methadone opioids. The researchers conclude that MMT is associated with subsequent MVC.

The manuscript has a number of strengths. First, the researchers aim to examine an important question that has implications for clinicians, patients, family members and motor transport administrators. Second, the use of administrative data can generate large sample sizes and might make results more reflective of real-world experience. Third, use of police-reported crash data can avoid the social desirability bias that afflicts studies based on self-report.

However, the manuscript also has a number of major limitations:

– The unusual organization and writing style of the manuscript made it difficult for me to understand the methods and results, which in turn made it difficult to assess the manuscript strengths and weaknesses. The writing requires substantial revision before readers could reasonably be expected to fairly assess the data that is presented. In terms of organization, I was surprized to see that "Materials and methods" followed "Discussion".

– What population were the new MMT users drawn from? Figure 1 suggests these individuals are justice system data rather than from medical data. Were these individuals compelled to undertake MMT as part of a criminal conviction? It was surprizing that there were only 1012 individuals with new MMT starts over a period of 7 years in a country of 23 million people. Indeed, the box for the SMT database in Figure 1 suggests almost 36,000 individuals started MMT between 2007 and 2016. A more fulsome discussion of the data sets and their inclusion criteria might help readers understand these issues better.

– Similarly, who were the control individuals? Were these new users of *prescription* non-methadone opioids? Did they have acute pain (e.g. after a surgery) or chronic pain (e.g. osteoarthritis)? Were the non-methadone opioids prescribed as opioid agonist therapy (e.g. MS Contin, suboxone)? Were they *illicit* opioids? If there was a medical indication for opioid use, why were these individuals in data sets from the justice system? Readers will need to understand the control group better in order to interpret the results.

– As mentioned above, I did not understand how the index date for the control group was established. The manuscript would benefit from a clearer explanation of this point.

– A major weakness of analyses that use administrative data is the lack of information on road exposure (hours or distance driven in a given period). Unobservable differences in road exposure can confound the relationship of interest. For example, if the control group contains many people prescribed opioids after major surgery, it is plausible that many of these individuals would not be driving and therefore not at risk of crash as they recovered from surgery. They would therefore be an unfair control group. This reiterates the importance of understanding the sampling frame from which the control group is drawn.

– The authors used a proportional hazards model, but their results suggest the proportional hazards assumption was violated. Another method might be more appropriate.

– How did researchers build their model? Why did they include MVC history, history of alcohol-impaired driving, antidepressant use and Z-drug use in their final model if they were not significant on adjusted analysis? Is there a reason to believe these are confounders?

My main points and suggestions are captured above. However, I would like to encourage the researchers to publish their work after more clearly describing their methods and potentially after revising their analysis. The topic is interesting, timely and important. Their data sources are extensive and easily linked. I think a substantially revised manuscript might contribute substantially to the literature on this topic.

[Editors' note: further revisions were suggested prior to acceptance, as described below.]

Thank you for resubmitting your work entitled "Risk of motor vehicle accidents after methadone use" for further consideration by *eLife*. Your revised article has been reviewed by 2 peer reviewers, one of whom is a member of our Board of Reviewing Editors, and the evaluation has been overseen by Timothy Behrens as the Senior Editor.

We have made some further suggestions, but these are entirely at your discretion. We are happy to publish the manuscript in its current state if you prefer:

In preparing your final version I would ask you to address the points about the MVA abbreviation and the SMT/MMT labels made by Reviewer #2. It might be advised to double check the numbers mentioned in comments 5 and 6 from Reviewer #2 as well.

Summary:

In the context of the ongoing opioid use crisis in many countries around the world, and the growing efforts to treat users with methadone maintenance therapy (MMT), it is critical to understand the risks and benefits of such therapies. This study addresses motor vehicle accident rates in opioid users in MMT as compared with the general adult population but also includes assessment of opioid users who are not on MMT. This inclusion is a key advance of understanding the risks of MMT. The primary limitation lies in the lack of precision and ability to address certain questions, due to the data available in the databases. This is balanced by the scope of the sample that is permitted by those databases. These results potentially contribute to reduce public harm by suggesting increased caution for MMT patients in the several months after initiation of therapy.

*Reviewer #1:*

Tsai and colleagues report a study of the rate of motor vehicle collisions in populations of opioid users on methadone maintenance therapy (MMT) versus those not on MMT and matched control non-opioid users. In the context of the ongoing opioid use crisis in many countries around the world, and the growing efforts to treat users with methadone maintenance therapy (MMT), it is critical to understand the risks and benefits of such therapies. This study addresses motor vehicle accident rates in opioid users in MMT as compared with the general adult population but also includes assessment of opioid users who are not on MMT. This inclusion is a key advance of understanding the risks of MMT. The study captures national statistics in Taiwan for a 7 year period (2009-2016) which include 3036 opioid users of which 1012 were in MMT. The first approach was to compare the incidence of motor vehicle collisions in each population in each year of the data interval. These data show the highest rates in MMT users and the lowest in the general population for each year of study. Most importantly, by tracking individuals after an index date of the first MMT dose in this group, and a matched date for the non-MMT controls, this study shows the cumulative probability of a motor vehicle accident increases more rapidly in the MMT opioid group versus the non-MMT opioid group in the first 100 days after the index date.

Major strengths of this study include the capture of a large number of individuals through national databases, the comparison of MMT and non-MMT opioid users groups and the determination of the latency of motor vehicle collisions after the initiation of therapy. Minor strengths are captured by the inclusion of covariates such as a prior history of motor vehicle collisions, driving under the influence and benzodiazepine use.

One limitation to the conclusions from this study lies in a lack of clarity about the opioid use history of the subjects as the approach only captures their first interaction with law enforcement. In addition, there is no indication of the factors in Taiwan that determine whether an opioid users is placed on MMT or not, particularly with respect to opioid use intensity and longevity of the pattern. The Discussion addresses this as necessary caveats due to the data captured in the available databases. The study has largely achieved the aims of the authors by investigating the risks posed by MMT for motor vehicle collisions. The results presented confirm increased risk and show that it develops over the first 100 days. The lack of precision and ability to address certain questions, due to the data available in the databases, is balanced by the scope of the sample that is permitted by those databases. These results potentially contribute to reduce public harm by suggesting increased caution for MMT patients in the several months after initiation of therapy.

*Reviewer #2:*

1. The old abbreviation MVAs is wrong because it is not equal to the current motor vehicle collision item! motor vehicle accident (MVA)

2. The conclusion continues to be too arbitrary, because the study suggests that we should prohibit from driving during MMT. However, in actual clinical practice, driving during heroin use or in a heroin withdrawal state may be more dangerous. The retrospective data analysis could not explain the patient severity or patient's condition. In general concept, patient with severe opiate use disorder (OUD) need MMT than mild or moderate opiate users, and the occasional/ or mild opiate users do not require MMT. Therefore, the severity of OUD may be the main factor affecting motor vehicle accident (collision) as in the figure 2 and 3.

3. The method remain unclear such as what reason that the punitive administrative system for the use of Category 3 or 4 Narcotics only. what this study do not used the Category 1 or 2? and Which category of narcotics does opiate belong to in Taiwan?

4. In figure 1, the Substitution maintenance therapy (SMT) is same the Methadone maintenance treatment (MMT) ?

5. The case number is unclear such as the new registry of opiate users between 2010 and 2016 (N=15996), but the new registry of opiate users with MMT (N=1145) and without MMT (N=14158), and 1145+14158 unequal to 15996. The opiate number in management information system of SMT have 35927 peoples from 2007-2016, but 2010-2016 only remain 1145, the proportion and frequency are inappropriate.

6. The number of exclusion criteria (N-153, and N=540) is also unclear, during Patient with OUD received SMT (2007-2016, N=35927) in figure 1. These data will confuse the readers.

7. The Ethics Statement that the study was approved by the Institutional Review Boards (TSMH IRB No.: 17-010-B1 and 18-049-B), but the statement shows Important flaws/ Disadvantage, because the IRB from TSMH IRB (TSMH is a local hospital, this institute have not FERCAP and AAHRPP approved) and all author affiliations do not from this hospital. However, the corresponding authors affiliations such as National Taiwan University and Kaohsiung Medical University IRB were approved by FERCAP and AAHRPP. the three corresponding authors affiliations (Jui-Hsiu Tsai affiliation: Dalin Tzu Chi Hospital/ Buddhist Tzu Chi Medical Foundation, Hung-Yi Chuang affiliation: Kaohsiung Medical University, Yue Leon Guo affiliation: National Taiwan University, and first author affiliations (Ya-Hui Yang) : Fooyin University, Kaohsiung

8. Data were retrieved from NHRID, and 6 Taiwanese population-based administrative registries (such as Ministry of Health and Welfare, the road accident registry of injurious crashes, National Police Agency, Ministry of the Interior, and Ministry of Justice), this scientific study should provide the approved document from these 6 population-based administrative registries. In addition, the NHRID do not the data of National Police Agency, Ministry of the Interior, and Ministry of Justice. Therefore, TSMH IRB No.: 17-010-B1 and 18-049-B should provide the approved document from NHRID, and from 6 Taiwanese population-based administrative registries for this local TSMH IRB.

---

## [Author Response]

Reviewer #1:[…] 1. Results, Page 11: The final paragraph is confusing, perhaps because it is not well described in the methods or the results that this analysis (and that in the lower panel of Figure 3) focuses on the sub-populations who did eventually get in a motor vehicle accident. This is not directly stated anywhere so if this is not true, then it is even less clear what is being depicted in Figure 3.

Thank you for your comment.

We added and revised the final paragraph on Pages 16-17 and further revised our Figure 3. See it in detail on Lines the last 1-7, Page 16 and Lines 1-6, Page 17.

“Figure 3 shows the Kaplan–Meier analysis for motor vehicle collision-free survival between opiate users receiving MMT and those not receiving in long-term and short-term follow-up. […] Overall, descending rate of motor vehicle collision-free survival in the MMT group (6.5%; range 5.23% to 7.08%; 95% confidence interval [95% CI], 5.23 to 7.08) was significantly higher than that in the control group (2.2%; range 1.90% to 2.41%; log-rank test p<0.001)(Figure 3).”

2. Figure 3: It is unclear from the legend and the labels on the axes what is being shown in the upper panel of Figure 3. The legend should be revised to make this very clear.

This is good suggestion. We also considered the upper panel of Figure 3 not clear. But our project was finished in 2018 and the database of our project was only used until 2020. So we couldn’t further draw the new figure; hence, we only reduced Y-axis scale from 0.5 to 1.0 in the upper panel of Figure 3. See it in detail on Figure 3.

3. Discussion Page 13: "this rate declined drastically…during the first 90 days." This appears to be referring to the lower panel of Figure 3 in which the cumulative rate of motor vehicle accidents increased over the first 90 days. This all needs to be better clarified, particularly with respect to the year intervals reported in Figure 3, upper panel.

We revised this sentence on Lines the last 4-7, Page 18.

“In the 7-year cohort study with frequency-matched controls, the motor vehicle collision incidence rate among opiate users was significantly higher in those receiving MMT than in those not receiving MMT. This rate drastically increased in opiate users receiving MMT during the first 90 days of follow-up. Rural area residents had increased motor vehicle collision risk. Although the occurrences of motor vehicle collisions in drivers under the influence of …”…

4. Discussion Page 14: It appears from the Introduction comment that the "3 observational studies" cited did not feature a non-MMT group with an opioid use history? If that is the case it should be re-emphasized here since this appears to be a strength of this manuscript.

Yes, the 3 observational studies only showed different risks of motor vehicle collisions between two populations (the MMT population and the general population). Additionally, we also added the opiate-using population in the preliminary study on Figure 2 and we should eliminate interfering factor of an opioid use history in the cohort study of opiate users receiving and not receiving MMT. So we added re-emphasized sentences in our “strengths and limitation of this study” section on Lines 1-6, Page 22.

“…, particularly in ethnic Chinese populations. Most observational studies showed different risks of motor vehicle collisions between two populations (the general population and the MMT population). We not only added the third population (the opiate-using population) in the preliminary study not also eliminated the key interfering factor of opioid use in the cohort study of opiate users receiving MMT and those not receiving.”

5. Discussion Page 16: Although it is appreciated that road conditions and driver behavior are not captured in the database, it may be that the time of day and day of the week when accidents occurred would be useful variables, and should be assessed.

Thank you for your suggestion.

Now, we couldn’t obtain the information about the time of day and day of the week when accidents occurred in our study. This is because our project was finished in 2018 and the database of the project was only used until 2020. But we further searched for related information in Taiwan. According to the statistics of the Highway Bureau in Taiwan, the most common time for fatal car accidents (34.0%) was between 0-6 A.M. and that for motor vehicle collisions (28.8%) was between 4-7 P.M. by the year 2017. (https://www.freeway.gov.tw/UserFiles/106%E5%B9%B4%E5%9C%8B%E9%81%93%E4%BA%8B%E6%95%85%E6%AA%A2%E8%A8%8E%E5%88%86%E6%9E%90(%E5%A4%96%E9%83%A8%E7%89%88%E6%9C%AC)(4).pdf).

6. Methods, Page 19: The authors should address the factors in Taiwan that determine whether an opioid users is placed on MMT or not, particularly with respect to opioid use intensity and longevity of the pattern.

We approved that it is an important public issue in Taiwan, even in the world; hence, we did not focus its issue in our study. According to our clinical observation and our view, MMT could effectively reduce occurrences of HIV in opiate users in Taiwan; and it might decrease opioid use intensity and longevity of the pattern in Taiwan. However, the results of our study were unable to support whether opioid use is placed on MMT or not in Taiwan. In future, it is essential for more studies in Taiwan to support its key issue.

7. Methods, Page 19: The selection of the "index date" for the non-MMT needs to be better described. Would it be possible to include an entry in the Table to describe the latency between the placement on the opioid registry and the index date for each group?

(1) We further clearly described the index data for the non-MMT group as possible as we could. See it in detail on Lines last 2-3, Page 12 and we revised the sentence as below:

“… the index date. Other new opiate users not receiving MMT were randomly selected as the unexposed (MMT) group after they were frequency-matched to the exposed group at a ratio of 1:2 according to age, sex, and opiate use duration. Thereafter, the index date of 2 matched unexposed users was the same day as that of the exposed user.”…

(2) Now, it is impossible for us to obtain these data about the latency between the placement on the opioid registry and the index date for each group because our project was finished in 2018 and the database of our project was only used until 2020.

8. Methods, Page 19: It should be stated directly that the subject was the driver of the car in the motor vehicle accidents, to avoid any confusion.

We want to state directly that and obtain the data about driver licenses. But our database was lacked. So we did not the subject to be the driver of the car.

9. Methods, Page 19: "on the basis of age, sex, and opiate use duration". This appears to suggest a design in which the non-MMT users were selected by matching to the MMT users but this needs to be better clarified.

Thank you for your suggestion. We tried to write those as clearly as we can. See it in detail on Lines last 3-6, Page 12. We revised as below:

“… the index date. Other new opiate users not receiving MMT were randomly selected as the (MMT) unexposed group after they were frequency-matched to the exposed group at a ratio of 1:2 according to age, sex, and opiate use duration. Thereafter, the index date of 2 matched unexposed users was the same day as that of the exposed user.”…

10. Methods, Page 18-19: It appears to be the case that all opioid users would have to have a conviction for use of illicit drugs. This should be clarified, particularly with respect to the description as "new" opioid users. Is this defined by their first interaction with / detection by law enforcement? Or was there some sort of interview to assess when they started using opioids?

In our study, new opioid users were defined the first detection by law enforcement because we used the data retrieved from the management information system of substitution maintenance therapy, Ministry of Health and Welfare and 4 independent management information systems at the Ministry of Justice, Republic of China (Taiwan). So we added a sentence in our text on Lines 8-9, Page 12. And our study did not use by any interview to assess.

“From the 4 registry databases, we identified new opiate users between 2010 and 2016 (n=15,996), who were defined the first detection by law enforcement. The new opiate users were excluded if they (1) …”

Reviewer #2:1) Data were retrieved from NHRID, and 6 Taiwanese population-based administrative registries, but the 6 population-based administrative registries may high repeat and overlapping such as 1. management information system of substitution maintenance therapy belongs to Ministry of Health and Welfare, 2. National Police Agency belong to Ministry of the Interior, 3. Should be clarify the MVA information of road accident registry of injurious crashes from Ministry of Transport or National Police Agency.

Thank you for your suggestion. Yes, these registries were high repeat and overlapping. So we used the unique national identification numbers assigned to each citizen in Taiwan to link among the different registries; then, we further need to clean data. So we described the MVA information of road accident registry of injurious crashes on Lines 7-10, Page 13.

2) The Methadone is a long action medicine, which induce motor vehicle accident less than patient with current OUD/or current OUD during opiate withdrawal, and less than current using BZD or alcohol. This study has not clear clarification these influence factors.

We agreed with your comment. The occurrences of motor vehicle collisions in drivers under MMT are less than those under the influence of current OUD, current OUD during opiate withdrawal, BZD, or alcohol. Those factors may be potential risk factors in driver under MMT. So we included these influence factors for motor vehicle collisions in MMT users. But we lacked for the data about driving under the influence of drugs such as BZD, OUD, or current OUD during opiate withdrawal. So we only used the history of drug use as influence factors in their index date. See it on Table 1. And we added some sentences about the top on Lines the last 1-3, Page18 and Page 19.

3. The study conclusion is too arbitrary, and suggested that receiving MMT have high motor vehicle accident (MVA) risk than not receiving MMT in patient with opioid use disorder (OUD), and should prohibition against driving during MMT.

Thank you. We revised these sentences of our study conclusion in the “Abstract” section on Lines the last 1-3 of Page 4.

“In conclusion, drivers receiving MMT have higher motor vehicle collision risk than those not receiving MMT in opiate users and we should prohibit from driving during MMT, particularly during the first 90 days of MMT.”

The major risk of driving is drinking driver and drug (opiate and BZD) driver, the study could not strong to excluded that multi-factory risks in MVA.

The major risks of driving are drinking driver and drugs (BZD and antidepressants). Due to lack of data about driving under the influence of the drugs, we used the past history of drinking driver, antidepressant, and BZD exposed as potential covariates in their index dates. You can see these covariates on Table 2 and Study population of the “Materials and methods” section on Lines 1-3, Page 13.

“Potential covariates, including history of motor vehicle collisions, driving under the influence (DUI), antidepressant use, and BZD (including Z-drug) use before the index date, were included in the analysis (…”

The treatment with MMT play an important successful role for control HIV infection in Taiwan patient with OUD, and OUD patient must regular take methadone every day.

Yes. In Taiwan, MMT plays an important successful role for control HIV infection in patients with OUD, and they must regularly take methadone every day.

4. Opiate users not receiving MMT may be just discharged from jail or enrolled in a NGO institution for rehabilitation, or their current severity of opiate use may not met the OUD criteria, or the current severity in non-MMT group may less than receiving MMT group (because received MMT case must met the criteria of OUD, and most case presents in severe degree). In addition, persistent opiate user may more high risk of MVA and other accident if not receiving any treatment, because persistent current opiate (heroin) user shows high craving for drug, or presents in withdrawal state. The high craving ad withdrawal state are all dangerous state for anything (including MVA and many harm accident).

Thank you for your comments. In our study, the data of opiate users were retrieved from the management information system of substitution maintenance therapy, Ministry of Health and Welfare, the road accident registry of injurious crashes, National Police Agency, Ministry of the Interior, and 4 independent management information systems at the Ministry of Justice. So we only obtained the information about opiate users met the heroin or opioid abuse criteria, but not the OUD criteria; and we exclude opiate users in jail after their index date (Figure 1). The data about enrollment in a NGO institution for rehabilitation and severity of opiate use were lacked. We wrote these limitations in the Strengths and limitations of this study section on Lines the last 1-3, Page 22.

“Moreover, opioid doses were recognised much higher risks among MMT users than among non-MMT opioid users; hence, opiate dose was uncaptured because it was unavailable in our database. The data about enrollment in a non-governmental organization institution for rehabilitation and severity of opiate use were lacked. Third, data from drivers’ licences …”

5. The Methadone is a long action medicine, which induce motor vehicle accident less than patient with current OUD/or current OUD during opiate withdrawal, and less than current using BZD or alcohol. These factors should clarification in the conclusion.

Thank you for calling attention to this opioid. We agreed with your opinion. The occurrences of motor vehicle collisions in drivers under the influence of current OUD, current OUD during opiate withdrawal, BZD, or alcohol are higher than those receiving MMT. Hence, we focused on the risk of motor vehicle collisions in opiate-use drivers comparing between receiving MMT and not receiving MMT. So we could include the history of opiate use, BZD, and alcohol exposure as the potential risk factors in the population because we lacked the information about driving under the influence of current OUD, current OUD during opiate withdrawal, BZD, or alcohol in our database. We added some sentences in the Discussion section on Lines the last 1-3, Page 18 and Page 19.

“... residents had increased motor vehicle collision risk. Although the occurrences of motor vehicle collisions in drivers under the influence of current opiate use, current opiate use during opiate withdrawal, BZD, and alcohol are higher than those receiving MMT, these factors (such as the history of BZD, antidepressants, and alcohol exposed) were non-significantly increased risk of motor vehicle collisions in opiate users receiving MMT.”

6. The data of UNODC using 2018 is too old, the UNODC data of 2020 is available.

Thank you for supplying us for news. We revised the UNODC data of 2018 to that of 2020. See it below, Lines 1-3, Page 8, and a new reference on Lines 9-10, Page 34.

**“**Introduction

Approximately 58 million people worldwide had opioid user in 2019, with 30 million accounting for opiate users (…”

Method:7. Data were retrieved from NHRID, and 6 Taiwanese population-based administrative registries, but the 6 population-based administrative registries may high repeat and overlapping such as 1. management information system of substitution maintenance therapy belongs to Ministry of Health and Welfare, 2. National Police Agency belong to Ministry of the Interior, 3. Should be clarify the MVA information of road accident registry of injurious crashes from Ministry of Transport or National Police Agency.

Surely, the high repeat and overlapping problems existed in our primary data. So we used personalized unique national identification numbers to link among these different systematic data; meanwhile, we clarified the date of their accidents or treatment in the observation cases from these data.

8. The Methadone is a long action medicine, which induce motor vehicle accident less than patient with current OUD/or current OUD during opiate withdrawal, and less than current using BZD or alcohol. This study has not clear clarification these influence factors.

We agreed with your points. Those factors may be potential risk factors in driver under MMT. So we included these influence factors for motor vehicle collisions in MMT users. But we lacked for the data about driving under the influence of drugs such as BZD, OUD, or current OUD during opiate withdrawal. So we only used the history of drug use as influence factors in their index date. See it on Table 1.

Reviewer #3:As I understand the study methods, the researchers used six linked Taiwanese national administrative datasets to identify 1012 adults initiating MMT between 2009 and 2016, matching these individuals on the basis of age, sex, and 'opiate use duration' to a control group of 2024 new users of non-methadone opioids. Individuals were eligible for inclusion if they were new users of opioids between 2010 and 2016. Individuals were excluded if they: were registered as new opioid users prior to age 20 years; used opioids prior to 2010 (I believe the washout period was 2007 – 2009); had incomplete information on age, sex, or "multiplicity" (I did not understand the final term and wonder if the authors intended to use another word); or if they were subsequently incarcerated.

(1) The "multiplicity" included education status, income, residing in area, etc. So we revised the sentence on Lines 11-12, Page 12.

“… if they (1) were registered at < 20 years of age; (2) used opiate before 2010; and (3) had incomplete information on age, sex, education status, income, residing in area, etc. Of the new opiate users,…”.

(2) We excluded study subjects if they were subsequently incarcerated after their index date. See it in detail on the low-right side of Figure 1.

The index date for the MMT group was the date of first MMT administration. I had difficulty understanding the index date for the control group. The primary outcome of interest was subsequent police-reported motor vehicle collision (MVC). Individuals were followed until MVC, death, end of follow-up in registry records, or end of study interval.

We revised and wrote clearly the sentence about the index date for the control group. Thereafter, the index date of 2 matched unexposed (MMT) users was the same day as that of the exposed (MMT) user. See it in detail on Lines the last 2-3, Page 12.

“ Other new opiate users not receiving MMT were randomly selected as the unexposed (MMT) group after they were frequency-matched to the exposed group at a ratio of 1:2 according to age, sex, and opiate use duration. Thereafter, the index date of 2 matched unexposed users was the same day as that of the exposed user.”.…

Using a proportional hazards model to assess MVC-free survival, researchers report that the hazards of MVC were significantly higher among MMT users than among controls (crude hazard ratio, 3.0). After adjustment for income level, urbanity, education status, history of motor vehicle accidents, prior alcohol-impaired driving, prescription Z-drug exposure, and antidepressant use, researchers report that the adjusted hazard ratio for MVC was 2.8-fold higher among individuals receiving MMT relative to individuals receiving non-methadone opioids. The researchers conclude that MMT is associated with subsequent MVC.The manuscript has a number of strengths. First, the researchers aim to examine an important question that has implications for clinicians, patients, family members and motor transport administrators. Second, the use of administrative data can generate large sample sizes and might make results more reflective of real-world experience. Third, use of police-reported crash data can avoid the social desirability bias that afflicts studies based on self-report.However, the manuscript also has a number of major limitations:– The unusual organization and writing style of the manuscript made it difficult for me to understand the methods and results, which in turn made it difficult to assess the manuscript strengths and weaknesses. The writing requires substantial revision before readers could reasonably be expected to fairly assess the data that is presented. In terms of organization, I was surprized to see that "Materials and methods" followed "Discussion".

OK, thank you for your note. The main text of the article in the journal *eLife* should usually be structured and ordered as follows: Introduction; Results; Discussion; Materials and methods (or Methods); Acknowledgements; References; Figures with the corresponding legend below each one; and Tables. A Methods or Model section can appear after the Introduction where it makes sense to do so.

So we revised and changed the “Materials and methods” after the Introduction. See it in detail in our text.

– What population were the new MMT users drawn from?

In the present study, the new MMT users were new opiate users receiving new MMT (Figure 1). So the population drawn from the 5 Taiwanese population-based administrative registries, including 4 justice system data and 1 medical data.

Figure 1 suggests these individuals are justice system data rather than from medical data. Were these individuals compelled to undertake MMT as part of a criminal conviction?

No, the data of MMT users was mainly retired from the medical data and we also excluded the MMT users in jail.

It was surprizing that there were only 1012 individuals with new MMT starts over a period of 7 years in a country of 23 million people. Indeed, the box for the SMT database in Figure 1 suggests almost 36,000 individuals started MMT between 2007 and 2016. A more fulsome discussion of the data sets and their inclusion criteria might help readers understand these issues better.

Almost 36,000 MMT users were collected only 1012 new opiate users receiving MMT. This is because we collected new opiate users receiving the first MMT; meanwhile, they needed to match with the controls. So the study cases decreased much.

– Similarly, who were the control individuals?

The control individuals were new opiate users not receiving MMT between 2010 and 2016; meanwhile, they were frequency-matched to the cases according to age, sex, and opiate use duration. See it detail on Lines 6-15, Page 11.

Were these new users of prescription non-methadone opioids?

No, we used the 4 justice system data and new users almost used heroin and few other opioids.

Did they have acute pain (e.g. after a surgery) or chronic pain (e.g. osteoarthritis)?

No, we didn’t know whether they have acute pain or chronic pain in our database because we used the 4 justice system data, but not medical data.

Were the non-methadone opioids prescribed as opioid agonist therapy (e.g. MS Contin, suboxone)?

No, this is because the non-methadone opioids prescribed as opioid agonist therapy were not existed before 2018.

Were they illicit opioids?

Yes, they used illicit opioids because the information sourced from the 4 justice system data. So we had no information about iatrogenic opioid users in our database.

If there was a medical indication for opioid use, why were these individuals in data sets from the justice system? Readers will need to understand the control group better in order to interpret the results.– As mentioned above, I did not understand how the index date for the control group was established. The manuscript would benefit from a clearer explanation of this point.

The index date for the control group was established. We revised our methods clear as possible as we can. See it in detail on Page 12.

“Of the new opiate users, we selected those receiving MMT as the (MMT) exposed group, who were regular methadone users. […] Thereafter, the index date of 2 matched unexposed users was the same day as that of the exposed user. The included participants had not been in jail after their index date.”.…

– A major weakness of analyses that use administrative data is the lack of information on road exposure (hours or distance driven in a given period). Unobservable differences in road exposure can confound the relationship of interest. For example, if the control group contains many people prescribed opioids after major surgery, it is plausible that many of these individuals would not be driving and therefore not at risk of crash as they recovered from surgery. They would therefore be an unfair control group. This reiterates the importance of understanding the sampling frame from which the control group is drawn.

Thank you for your good suggestions. We added the sentence in our text on Lines 8-11 Page 22.

“Nevertheless, we acknowledge several limitations. […] Motor vehicle collisions in the study were assessed using the police registry.”

– The authors used a proportional hazards model, but their results suggest the proportional hazards assumption was violated. Another method might be more appropriate.

Thank you for your suggestion. As far as we know, it might be suitable for the present study to use a proportional hazard model because we only used the multiple database and this study needs to be presented in chronological order. Hence, our project was finished in 2018 and the database of the project was only used until 2020. Please further supply us for your suggestions.

– How did researchers build their model? Why did they include MVC history, history of alcohol-impaired driving, antidepressant use and Z-drug use in their final model if they were not significant on adjusted analysis? Is there a reason to believe these are confounders?

The model of our study was built on 7 researchers and 2 reports; Of 9, 3 researchers were main idea from Bramness et al., 2012; Corsenac et al., 2012; Leveille et al., 1994. Because the risk factors for MVC in MMT users were scant, we collected recognized risk factors for MVC in drivers as potential confounders. See the references on Lines 1-3 Page 13 and Pages 27-29.

3 main references, see in below:

Bramness JG, Skurtveit S, Mørland J, Engeland A. 2012. An increased risk of motor vehicle accidents after prescription of methadone. Addiction 107:967–972. DOI: https://doi.org/10.1111/j.1360-0443.2011.03745.x, PMID: 22151376

Corsenac P, Lagarde E, Gadegbeku B, Delorme B, Tricotel A, Castot A, Moore N, Philip P, Laumon B, Orriols, L. 2012. Road traffic crashes and prescribed methadone and buprenorphine: a french registry-based case-control study. Drug and Alcohol Dependence 123:91–97. DOI: https://doi.org/10.1016/j.drugalcdep.2011.10.022, PMID: 22104480

Leveille SG, Buchner DM, Koepsell TD, McCloskey LW, Wolf ME, Wagner EH. 1994. Psychoactive medications and injurious motor vehicle collisions involving older drivers. Epidemiology 5:591–598. DOI: https://doi.org/10.1097/00001648-199411000-00006, PMID: 7841240

My main points and suggestions are captured above. However, I would like to encourage the researchers to publish their work after more clearly describing their methods and potentially after revising their analysis. The topic is interesting, timely and important. Their data sources are extensive and easily linked. I think a substantially revised manuscript might contribute substantially to the literature on this topic.

Thank you. We clearly revised and described our methods as possible as we could. So we first revised and changed the “Materials and methods” after the Introduction. And we tried to clearly revised and described our cases and controls in the methods on Page 12.

[Editors' note: further revisions were suggested prior to acceptance, as described below.]

Reviewer #2:1. The old abbreviation MVAs is wrong because it is not equal to the current motor vehicle collision item! motor vehicle accident (MVA)

We revised old abbreviation “MVAs” to “motor vehicle collisions” on Line 10, Page 4.

“From 2009 to 2016, the crude incidence rate of MVAs motor vehicle collisions was the lowest in the general adult population, followed by that in opiate adults, and it was the highest in adults receiving MMT.”

2. The conclusion continues to be too arbitrary, because the study suggests that we should prohibit from driving during MMT. However, in actual clinical practice, driving during heroin use or in a heroin withdrawal state may be more dangerous. The retrospective data analysis could not explain the patient severity or patient's condition. In general concept, patient with severe opiate use disorder (OUD) need MMT than mild or moderate opiate users, and the occasional/ or mild opiate users do not require MMT. Therefore, the severity of OUD may be the main factor affecting motor vehicle accident (collision) as in the figure 2 and 3.

Thank you for your suggestion. We agreed with your opinion. We revised the conclusion on Lines the last 1-3, Page 4.

“In conclusion, drivers receiving MMT have higher motor vehicle collision risk than those not receiving MMT in opiate users and we should prohibit from driving during MMT pay more attention to road safety in such drivers, particularly during the first 90 days of MMT.”

3. The method remain unclear such as what reason that the punitive administrative system for the use of Category 3 or 4 Narcotics only. What this study do not used the Category 1 or 2? and Which category of narcotics does opiate belong to in Taiwan?

Before 2009 in Taiwan, there were all data about illicit drug uses in the processing system of criminal records (such as the case management system of Drug Prevention and Control Center, processing system of criminal records, criminal case system of drug case prosecutor briefed the transfer of information).

In 2009, the government in Taiwan began to implement the decriminalization policy for illicit drug uses. Therefore, illicit drugs were divided into 4 categories according to their pharmaceutical ingredient and addiction. So the abusers with Categories 3 or 4 Narcotics were decriminalized, but they still needed to be treated or abstained.

After 2009, the data about Categories 1 and 2 Narcotics still existed in the processing system of criminal records, but the data about Category 3 or 4 Narcotics independently excluded in the system.

(1) Although Category 3 or 4 Narcotics included major related information about the users with barbiturates, BZDs, and their derivatives, we could obtain more information of opiate use because many drug abusers had multiple substance uses.

(2) The data about Category 1 or 2 belong to the processing system of criminal records (the case management system of Drug Prevention and Control Center, processing system of criminal records, criminal case system of drug case prosecutor briefed the transfer of information).

(3) Opiate use belongs to the use of Category 1 (almost) or 2 Narcotics in Taiwan because there are 4 Categories of Narcotics by “Narcotics Hazard Prevention Act” in Taiwan. See it in detail below (https://law.coj.gov.tw/ENG/LawClass/LawAll.aspx?pcode=C0000008):

Category 1 Narcotics include heroin, morphine, opium, cocaine, and their derivative products.

Category 2 Narcotics: opium poppy, coca, cannabis, amphetamine, pethidine, pentazocine, and their derivative products.

Category 3 Narcotics: secobarbital, amobarbital, nalophine, and their derivative products.

Category 4 Narcotics: alprazolam, diazepam, and their derivative products.

4. In figure 1, the Substitution maintenance therapy (SMT) is same the Methadone maintenance treatment (MMT) ?

Yes, SMT is the same as the MMT in our study. There were 2 substitution maintenance therapies in Taiwan, including (almost) methadone maintenance treatment (MMT) and (few) buprenorphine maintenance therapy. MTT is the most widely-used maintenance treatment in Taiwan because the government provided public expenses for medical treatment; hence, buprenorphine maintenance therapy is a self-financed medical treatment. So almost opiate users received MTT, but few those received buprenorphine maintenance treatments if they wanted (or needed by the law problems) to quit illicit drugs. Because Management information system of substitution maintenance therapy only collected the information about public medical treatment with MMT, our data could not include the information about buprenorphine maintenance therapy.

We added extra-explanation in figure 1. See it in detail in figure 1. “^a^ The system only collected the information about public fee for MMT.”

5. The case number is unclear such as the new registry of opiate users between 2010 and 2016 (N=15996), but the new registry of opiate users with MMT (N=1145) and without MMT (N=14158), and 1145+14158 unequal to 15996. The opiate number in management information system of SMT have 35927 peoples from 2007-2016, but 2010-2016 only remain 1145, the proportion and frequency are inappropriate.

We revised our Figure 1 in order to be more clear understanding our flowchart.

(1) The new registry of opiate users between 2010 and 2016 (N=15996), the new registry of opiate users with first MMT (N=1145), without MMT (N=14158), and exclusive subjects (N=153+540), and 1145+14158+153+540 is equal to 15996. See it in detail on Figure 1.

(2) A study in Taiwan showed that 19.86% of 3,343 opiate users received MMT in 2009 (Taiwan J Public Health. 2012;31(5):485-497; http://dx.doi.org/10.6288/TJPH2012-31-05-09). In this study, opiate users were all not new registry. So according to this study, we crudely estimated the new registry of opiate users with MMT: 15,996 X 19.85% = 3,199. In our study, there were 1,838 new registry of opiate users with MMT. 1,838/15,996 = 11.49%. That is because new registry of opiate users may be less willing to receive MMT than other opiate users in our study. So the number of our study subjects led to be obviously decreased.

6. The number of exclusion criteria (N-153, and N=540) is also unclear, during Patient with OUD received SMT (2007-2016, N=35927) in figure 1. These data will confuse the readers.

(1) N=153. The 153 subjects received SMT before they had opiate use at a registry. We had no information about how long they had opiate use before they received SMT. So we could not match the control.

(2) N=540. The 540 subjects went to jail after they had new registry of opiate use with MMT. Because they stayed to jail and had on risk of traffic accidents during the study period, we exclude them in our study.

7. The Ethics Statement that the study was approved by the Institutional Review Boards (TSMH IRB No.: 17-010-B1 and 18-049-B), but the statement shows Important flaws/ Disadvantage, because the IRB from TSMH IRB (TSMH is a local hospital, this institute have not FERCAP and AAHRPP approved) and all author affiliations do not from this hospital. However, the corresponding authors affiliations such as National Taiwan University and Kaohsiung Medical University IRB were approved by FERCAP and AAHRPP. the three corresponding authors affiliations (Jui-Hsiu Tsai affiliation: Dalin Tzu Chi Hospital/ Buddhist Tzu Chi Medical Foundation, Hung-Yi Chuang affiliation: Kaohsiung Medical University, Yue Leon Guo affiliation: National Taiwan University, and first author affiliations (Ya-Hui Yang) : Fooyin University, Kaohsiung

Thank you for best suggestion. If we applied for the FERCAP and AAHRPP approved IRB, our statement should show the prefect flaws. Hence, we only applied for local IRB. This is because the present study originated from 2 research projects, which of π was Dr. Ya-Hui Yang and the projects were funded by Taiwan Food and and Drug Administration (MOHW106-FDA-D-114-000671 and MOHW107-FDA-D-114-000636). There was no special requirement for IRB, such as the IRB approved by FERCAP and AAHRPP, in the study of Taiwan Food and Drug Administration. Dr. Yang want to draw perfective projects so she applied for local IRB because she taught in a University. Meanwhile, we used databases which were the use of previously stored de-identified information. We should ensure protection of the rights and welfare of human research subjects. Before and after our study, we still kept the same risk-benefit for research subjects.

8. Data were retrieved from NHRID, and 6 Taiwanese population-based administrative registries (such as Ministry of Health and Welfare, the road accident registry of injurious crashes, National Police Agency, Ministry of the Interior, and Ministry of Justice), this scientific study should provide the approved document from these 6 population-based administrative registries. In addition, the NHRID do not the data of National Police Agency, Ministry of the Interior, and Ministry of Justice. Therefore, TSMH IRB No.: 17-010-B1 and 18-049-B should provide the approved document from NHRID, and from 6 Taiwanese population-based administrative registries for this local TSMH IRB.

Not all right.

Yes, the NHRID do not include the data of National Police Agency, Ministry of the Interior and Ministry of Justice.

No, only TSMH IRBs should not provide all document/data. See it in detail below:

Our projects were mainly supervised and provided the approved data by the 3 authorities, including Taiwan Food and Drug Administration (TFDA), a local hospital (TSMH IRBs), and the Health and Welfare Data Science Center, Ministry of Health and Welfare.

First, we needed to propose the research projects to TFDA.

After TFDA agreed with our projects (MOHW106-FDA-D-114-000671 and MOHW106-FDA-D-114-000636, pp. 24), TFDA helped us to obtain study information from different authorities like Ministry of Health and Welfare (NHIRD and the management information system of substitution maintenance therapy), National Police Agency, Ministry of the Interior (the road accident registry of injurious crashes), and Ministry of Justice (the 4 independent management information systems). These authorities imported related data into HWDC after they agreed with us.

Second, we applied and provided the document by a local hospital (TSMH IRB).

Third, we submitted the above 2 documents to the Health and Welfare Data Science Center (HWDC), Ministry of Health and Welfare (Line the last 1 Page 9 and Line 1 Page 10) for applying for use of these database. HWDC provide a research platform to connect different database data by unique national identification numbers (ID).

Finally, we performed data analysis and statistical analysis in the HWDC.

For the consideration of privacy protection, all of the personal identifications are recorded, only authorized researchers are permitted to process databases in a separated designate area, and only statistical results are allowed to be carried out for publications.

So we provided and quoted the paper because HWDC and NHRID were detailed in this paper (Line 2 Page 10 and Line the last 2-4 Page 29). You should see the Data availability section on Page 25.

We also added this sentence (below) in the Material and methods section on Line 4-7 Page 10.

“Data from different systems were linked using the unique national identification numbers assigned to each citizen in Taiwan. […] Personal identifiers were removed after the linkage and before the analysis.”